

# Adjoint-Based High Fidelity Concurrent Aerodynamic Design Optimization of Wind Turbine

Sagidolla Batay[1], Bagdaulet Kamalov[1], Dinmukhamed Zhangaskanov[1], Yong Zhao[1], Dongming Wei[2], Tongming Zhou[3], Xiaohui Su[4]

[1]Dept. of Mechanical & Aerospace Eng., School of Engineering and Digital Sciences, Nazarbayev University, Astana, 010000, Rep. of Kazakhstan
[2]Dept. of Mathematics, School of Humanities and Sciences, Nazarbayev University, Astana, 010000, Rep. of Kazakhstan
[3]Dept. of Civil Environmental and Mining Engineering, The University of Western Australia Perth, WA 6009, Australia
[4]Key Laboratory of Ocean Energy utilization and Energy Conversion of Ministry of Education, Dalian University of
Technology, Dalian 116024, China

Corresponding author, email: yong.zhao@nu.edu.kz

**Abstract.** The aim of this study is to contribute towards the development of an open source MDO (multidisciplinary design optimization) platform DAFoam (https://github.com/DAFoam) in general and develop, implement, and integrate various technologies for wind turbine MDO for the energy community in particular, and to implement hi-fidelity concurrent multi-
disciplinary the aerodynamic design optimization in terms of five different schemes as well as to contribute to the development of the opensource MDO software, DAFoam. To evaluate novel turbine designs, the wind energy sector extensively depends on computational fluid dynamics (CFD). To use CFD in the design optimization process, where lower-fidelity approaches like blade element momentum (BEM) are more popular, but new tools to increase the accuracy must be developed as the latest wind turbines are larger and aerodynamics and structural dynamics become more complex. In the
present study, a new concurrent aerodynamic shape optimization approach towards multidisciplinary design optimization (MDO) that uses a Reynolds-averaged Navier Stokes solver in conjunction with a numerical optimization methodology is introduced. A multidisciplinary design optimization tool called DAFoam is used for the NREL phase VI turbine as baseline geometry to conduct extensive aerodynamic design optimizations such as cross-sectional shape, pitch angle, twist, chord length, and dihedral optimization. Pointwise, a commercial mesh generator, is used to create the numerical meshes. As the
adjoint approach is strongly reliant on the mesh quality, up to 17.8 million mesh cells were employed during the mesh convergence and result validation processes, whereas 2.65 million mesh cells were used throughout the design optimization due to the computational cost. The nonlinear optimizer SNOPT is used for the optimization process in the adjoint solver. The torque in the tangential direction is the optimization's merit function and excellent results are achieved, which shows the promising prospect of applying this approach for transient MDO. This work represents the first attempt to implement
DAFoam for wind turbine aerodynamic design optimization.

**Key words:** High-fidelity optimization, Aerodynamic optimization, SNOPT, DAFoam, Concurrent design optimization, NREL phase VI



## 1. Introduction

Wind energy has been used as a source of energy by mankind for a long period of time prior to the industrial revolution. Since the dawn of industrialization, fossil fuels have played a significant role in supplying energy. As concerns about the fast depletion of fossil fuels and global warming have grown, investment and research in renewable energy sources, including wind energy, have surged significantly. Due to the potential for wind energy to be accessible worldwide and its sustainability, it is one of the most prominent alternatives to investigate and improve further in order to collect more energy from the wind. There are two major components in wind turbine blade design optimization: structural design optimization, aerodynamic design optimization, which requires aerodynamic shape design through numerical simulations or experiments. Above all, the aerodynamic design optimization design is critical, since it affects both the mechanical characteristics and the fluid-structural performance of the wind turbine blades.

Optimizing the blade design is critical for its improving aerodynamic and mechanical performance, as well as increasing the amount of energy collected by wind turbines. Though, due to the sophisticated nature of the aerodynamics and the interaction of the air with the blade structure, optimizing the blade design is the most difficult aspect of wind turbine optimization. As a result, advanced numerical simulation techniques must be used. Currently, wind turbine design optimization software such as Ansys, OpenFOAM, Fast, Open Fast, QBlade, PreComp and BModes, Flex 5 and Matlab are utilized (Mishnaevsky et al., 2017), while in certain research, XFoil, NACA airfoils, and AirfoilPrep are used to simulate the aerodynamics of the blades in conjunction with Matlab. To attain the desired lift and drag coefficients necessary to generate maximum power, determining the optimal angle of attack at each span section is critical, as are the shape of the airfoil, twist in the blade and thickness of the airfoil throughout the span.

One of the most effective methods of optimization is to divide the blades into multiple portions along their span and examine their overall aerodynamic qualities using numerical simulation techniques. Because the primary goal of current wind turbine blade design is to minimize the cost and weight and simultaneously increase the energy collection efficiency by the wind turbine with deformation constraints, multidisciplinary concurrent optimization of aerodynamic, structural, and fluid-structure interaction should be used to improve wind turbine performance (Eminoglu & Ayasun, 2014).

As one of the most critical components of a wind turbine, the turbine blade is critical for improving the power production which is also exposed to complex external aerodynamic stresses (Zheng et al., 2017). The optimization of the blade is divided into two branches: structural optimization and aerodynamic optimization (Burton et al., 2011). Aerodynamic optimization is concerned with the optimal aerodynamic design, loads and noise levels, as well as the produced power efficiency, while structural optimization is concerned with weight, mechanical performance such as bending rigidity/strengths and fatigue life span, and so on.

An ideal optimization procedure would include both branches as an optimization goal when numerous objectives are considered. However, intricate research settings and processes make it difficult for researchers to choose these branches rather than simplifying the procedure and optimizing two branches sequentially (Zhu et al., 2016). While the majority of



studies investigate either structural or aerodynamic optimization, others explore the combination of the two. In terms of aerodynamic optimization, most of the work has been done using BEM rather than CFD (see below for more details). As a result, high-fidelity aerodynamic design optimization is critical for optimizing wind turbines. The current study performed concurrent high-fidelity full shape optimization, planform design optimization, and pitch angle optimization, and produced and described the optimal solutions in detail.

## 2. Literature Review

### 2.1 Low-fidelity aerodynamic design optimization

Wind turbine performance (power production and loads) has been shown to be affected by the aerodynamic forces created by the wind. The aerodynamic design of a wind turbine rotor is the most critical aspect in wind turbine design. In general, it entails determining the rotor shape and forecasting the rotor's aerodynamic performance. Numerous computational methodologies, such as blade element momentum (BEM) theory, vortex wake method, and computational fluid dynamics, have been used to analyze wind turbines' aerodynamics (CFD) and their design performance. Due to its simplicity and accuracy, the BEM theory is frequently utilized in the aerodynamic design of wind turbines (Lanzafame & Messina, 2007). The BEM theory is based on Glauert's propeller theory (Southwell, 1927).

Robustness, on the other hand, has been a concern with BEM codes, since they do not always converge (Maniaci, 2011). Robustness is crucial, even more so when the analysis is performed as part of an optimization process. In the best-case scenario, a lack of resilience will cause the convergence to be slowed down, while in the worst-case scenario, the optimization will be terminated entirely. Ning (Andrew Ning, 2013) addressed this problem by re-parameterizing the BEM equations with a single local inflow angle, ensuring convergence. Overall, BEM-based optimization approaches provide excellent results at a low computational cost (Hansen, 2013)for certain flow conditions.

Numerous studies on wind turbine blade shape optimization have been conducted. Chehouri et al. (Chehouri et al., 2015) conducted an in-depth examination of wind turbine optimization strategies. Ceyhan (2008) optimized the aerodynamic performance of horizontal axis wind turbine blades using BEM theory and a genetic algorithm (GA) (Ceyhan, 2008). The distribution of chords and twist angles are treated as design factors and adjusted for maximum power generation.

In the BEM, the blade is assumed to be composed of multiple separate spanwise parts, where the induced velocity at each element is calculated by maintaining a momentum balance on an annular control volume encompassing the blade element. While this model is simple and straightforward to use, it is accurate enough for many flow conditions since it cannot precisely assess the influence of the wake and sophisticated three-dimensional flows due to its simplistic assumptions.

Kenway and Martins (Kenway & Martins, 2008)optimized blade shape using a multidisciplinary design feasible (MDF) technique. An output increase of 3–4% was found for a 5-kW wind turbine, illustrating the method's potential. Clifton Smith and Wood (Clifton-Smith & Wood, 2007) improved both the power coefficient and minimized the starting performance of





tiny wind turbine blades. Liu et al. (Liu et al., 2007)devised a strategy for optimizing wind turbine blades in order to optimize the yearly energy output.

## 2.2 Medium-fidelity aerodynamic design optimization

Vortex techniques with a moderate degree of realism are often used as aerodynamic models in wind turbine applications. Vortex theory is based on potential flow, which does not account for the viscous effects that RANS CFD does. However, it produces a more realistic solution than BEM codes while maintaining a lower computational cost than CFD. The GENeral Unsteady Vortex Particle (GENUVP) code (Voutsinas, 2006), the Aerodynamic Wind Turbine Simulation Module (AWSM) (Kim et al., 2020), and the method for interactive rotor aeroelastic simulations (MIRAS) (Villalpando et al., 2011) are all

well-established vortex algorithms in the wind energy sector. Numerous sophisticated vortex wake models, namely the lifting line (Afjeh & Keith, 1986) and lifting surface (Tescione et al., 2016) models, are utilized to quantify the wake-induced velocities, blade loads, and wake geometry, which are the three most critical problems in rotor performance analysis.

Another optimal design method has been established using the lifting line theory to tackle a vortex model with a limited

number of blades. It is an impenetrable way for dealing with the theory of the wind turbine blades. It is hypothesized that the wake shed along the blades forms a helicoidal vortex sheet emanating from the trailing edge. The system of free vortices emerging from the trailing edge is likewise predicted to exist indefinitely downstream of the rotor disc by following a local streamline (Ferziger et al., 2020). While these vortex programs have been routinely used in analysis, they have been less often applied to design optimization. And BEM is still well-established and is still used as the default optimization method.

The most sophisticated approaches are based on solving the Navier Stokes equations, which can account for 3D flow, viscous, turbulent and compressible effects. However, the computational expense of these approaches restricts their applicability to only the most critical instances requiring very exact answers (Lerbs, 1952).

## 2.3 High-fidelity Gradient-based aerodynamic design optimization

Aerodynamic shape optimization based on high-fidelity CFD simulations has been more essential in the field of design

optimization during the last two decades. However, it continues to face difficulties related with high computational costs, which may be prohibitively costly when doing a large number of computationally expensive CFD simulations. As a result, it is critical to create more efficient aerodynamic shape optimization algorithms that can achieve an ideal design with the fewest feasible high-fidelity and costly CFD simulations.

The known approaches for optimizing aerodynamic shapes are classed as gradient-based, gradient-free heuristic, and

surrogate-based optimization (SBO). Gradient-based methods are very efficient when the gradients are calculated using Jameson's adjoint technique (Jameson, 1988a). The disadvantage is that the optimality of the solution might be very dependent on the initial predictions, and it can get caught in a local minimum (Chernukhin & Zingg, 2013).





As an alternative to the gradient-free heuristic technique, metaheuristic optimization algorithms such as Genetic Algorithms (GA), Simulated Annealing (SA), or Particle Swarm Algorithm (PSA) have strong global optimization capabilities. When
used for aerodynamic shape optimization, however, this sort of technique often takes thousands of CFD simulations or perhaps more, and the whole computing cost may quickly surpass the allotted computational budget. As a result, their applications have often been limited to two-dimensional aerodynamic configurations or three-dimensional configurations employing low-fidelity and quick CFD modeling techniques.

Optimization Using Surrogates (SBO). (Simpson et al., 2001) is a class of optimization methods that use low-cost surrogate
models to mimic costly objective and constraint functions, directing the addition and assessment of fresh sample points toward the global optimum. Nonetheless, when the number of design variables increases, the computing cost of optimization increases fast and quickly becomes prohibitively expensive.

When an efficient gradient assessment is available, gradient-based approaches are preferred. Gradient-based optimization was pioneered in the 1970s with gradients generated using finite-difference approximations (Hicks & Henne, 1978). With an
increasing number of design variables, the cost of this calculation becomes too expensive. To overcome this problem, adjoint techniques were created which enable the evaluation of gradients at a cost that is independent of the number of design variables. Peter and Dwight (Peter & Dwight, 2010) discussed these and more techniques for calculating aerodynamic shape derivatives. Martins and Hwang (Martins & Hwang, 2013) expanded on the adjoint approach and studied its relationship with various methods of derivative assessment.

### 145    2.3.1 High-fidelity Gradient-based aerodynamic design optimization with Adjoint method

Gradient-based algorithms are the only option for design optimization with a large number of variables if one intends to achieve convergence to an optimum in a reasonable length of time (Yu et al., 2018). The performance of this method is highly reliant on the cost and precision of calculating the gradients. While finite differences are easy-to-implement methods for computing gradients, they are prone to numerical errors and scale poorly with the number of design variables (Martins et
al., 2003).

Due to the fact that high-fidelity aerodynamic shape optimization models are driven by a system of non-linear Partial Differential Equations (PDEs), performing an accurate differentiation of the system of PDEs is quite difficult. This is one of the reasons for the scarcity of results from high-fidelity gradient-based aerodynamic shape optimization.

Pironneau proposes an adjoint technique in fluid mechanics (Pironneau, 1973), and Jameson expanded it for aerodynamic
shape optimization (Jameson, 1988b). Since then, the adjoint approach has been extensively employed in gradient-based optimization for aerodynamics applications (Nielsen & Anderson, 1999).

There are two distinct techniques to construct the adjoint equations for a primal solver based on the partial differential equations (PDEs): continuous and discrete (Mavriplis, 2007). The continuous technique uses the adjoint formulation of the governing equations from the original ones and then discretizes them for numerical solution. Anderson and
Venkatakrishnan's early work (Anderson & Venkatakrishnan, 1999), as well as the earliest adjoint implementations in





OpenFOAM (Othmer, 2008), all used this method. Continuous adjoints are more efficient and need less memory than the discrete adjoints. However, the accuracy of the continuous adjoint method degrades on coarse meshes (Papoutsis-Kiachagias & Giannakoglou, 2016).

At the moment, there is little study on high-fidelity aerodynamic shape optimization of wind turbine blades. However, one

effort has been made to solve the NREL VI wind turbine blade using a RANS solver and a continuous adjoint technique (Economon et al., 2013). T. Dhert et al. (Dhert et al., 2017) and Mads H. Aa. Madsen et al. (Madsen et al., 2019) have recently published exceptional work on this topic. Dhert et al. (Dhert et al., 2017) employed a discrete adjoint solver to perform multipoint optimization on a two-bladed rotor with a 2.6 million cell mesh, maximizing the torque coefficient while restricting the pitch, twist, and local shape design factors. And Mads H. Aa. Madsen et al. (Madsen et al., 2019) employed a

discrete adjoint solver to perform multipoint optimization on a three-bladed rotor with a 14.16 million cell mesh, optimizing pitch, twist, and local shape, chord as design variables while restricting thickness, thrust, and flap-wise bending momentum. While the same method was used in both situations, Mads H. Aa. Madsen et al used a more memory-efficient reverse automatic differentiation.

Vorspel et al. (Vorspel et al., 2018) recently conducted an unconstrained optimization of the NREL Phase VI rotor by

altering up to nine twist design factors using a steepest descent optimization technique. They highlighted their concerns over the convergence, which is not surprising given that the turbine is stall-regulated and exhibits split flow at certain intake speeds. Vortices at the tip and root exacerbated the problem of convergence, resulting in poor gradient quality. They solved this problem by restricting the deformable region to the outside 50% of the blade length, hence limiting shape optimization.

Our current research builds on the previously described work and implements new improvement. The primary enhancement

is the addition of dihedral as a design variable for optimization, which was not previously considered in wind turbine research. The DAFoam software (He et al., 2018, 2020) is implemented and utilized in this investigation, and it is based on the open-source CFD solver OpenFOAM. To the authors' best knowledge This study represents the first ever work on wind turbine optimization implementing and using the DAFoam to date based on adjoint and RANS solvers with reverse AD technique.

**3.    Methodology and mathematical modeling**

As described above, the entire high-fidelity aerodynamic design optimization is performed using the opensource program DAFoam (He et al., 2018). The package is built on the Mach-aero framework (https://github.com/mdolab/MACH-Aero), which includes pyOptSparse (Wu et al., 2020) and pyGeo (Kenway et al., 2010) for optimization, as well as OpenFOAM as a fluid solver. The tool is based on the adjoint technique, a fast method for generating derivatives that enables gradient-based

optimization to be applied to the systems with a high number of design variables. Given the large number of design variables and the complexity of the design process, a commercial nonlinear optimizer called SNOPT (Gill et al., 2002) is used. This



section details the geometry, mesh generation method, CFD, verification, and validation, as well as the formulation for design optimization.

## 3.1 RANS formulations and CFD methods

### 3.1.1 Baseline geometry

The optimization is based on the NREL Phase VI geometry, which was created for applied CFD validation tests. The NREL Phase VI test is a full-scale Unsteady Aerodynamic Experiment (UAE) conducted in the NASA-Ames wind tunnel on the double-bladed 10.058 m diameter NREL Phase VI Rotor based on the S809 airfoil (Hand et al., 2001). Figure 1 illustrates the baseline geometry.

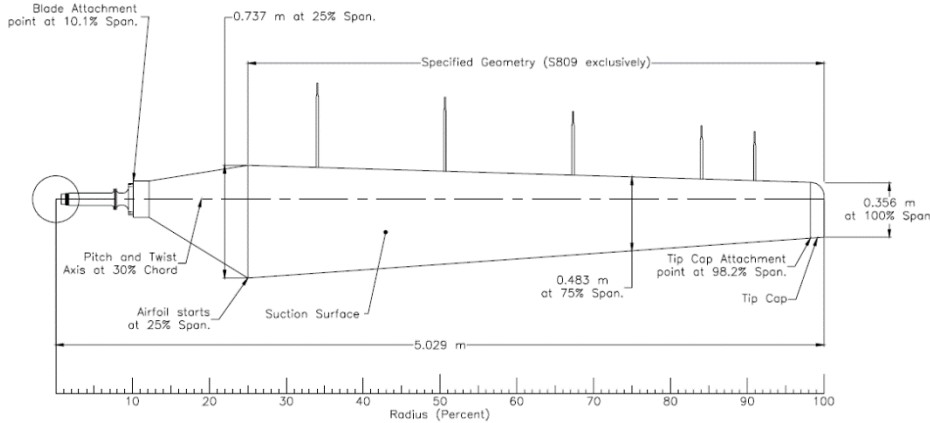

**Figure 1. Blade planform dimensions (Hand et al., 2001)**

The design process included rigorous trade-off analyses that examined nonlinear taper and twist distributions, as well as the incorporation of additional airfoils. The outcome is a blade with linear taper and a nonlinear twist distribution that utilizes the S809 airfoil from root to tip. This baseline geometry offers a suitable starting point for optimization while providing the potential for further performance gains.

NREL has quantified the knowledge on the 3D aerodynamic behavior of full-scale, horizontal-axis wind turbines (HAWT) utilizing the Phase VI wind turbine (Hand et al., 2001; Simms et al., 2001). The NREL Phase VI wind turbine has full-span pitch control and a 20-kW rating. The wind tunnel tests, conducted in 1999 at NASA's Ames Research Center, are widely regarded as a gold standard for evaluating wind turbine aerodynamic simulation.



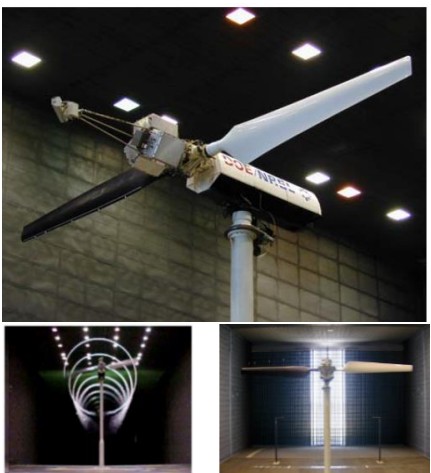

**Figure 2. NASA's Ames Research Centre/NREL Phase VI wind turbine (Hand et al., 2001)**

For wind speeds ranging from 5 to 25 m/s, a variety of experimental observations were made for blade surface pressures, integrated aerodynamic forces, shaft torque and thrust, blade root strain, tip acceleration, and wake. This turbine has two blades, a 12.2 m hub height, and a 10.058 m rotor diameter. The blade profile, which is linearly tapered and nonlinearly twisted, is based on the S809 airfoil form (Ramsay et al., 1995; Somers, 1997). The NREL study contains detailed information regarding the structural geometries, mechanical and material characteristics, as well as experimental findings and methodology (Hand et al., 2001; Simms et al., 2001). The trailing edge of the airfoil has been slightly modified to have a higher mesh quality in order to prevent severe non-orthogonality during the mesh generation.

### 3.1.2 3.1.2 Mesh generation and mesh convergence studies

The mesh domain is made up of blades and a fluid domain around them. A meshing program, Pointwise, was utilized to build hybrid meshes across the domain, including completely structural meshes around the blade and non-structural meshes in the farfield domain.

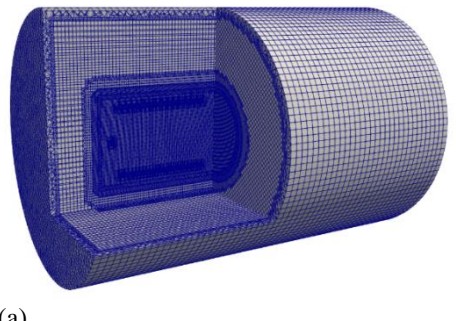

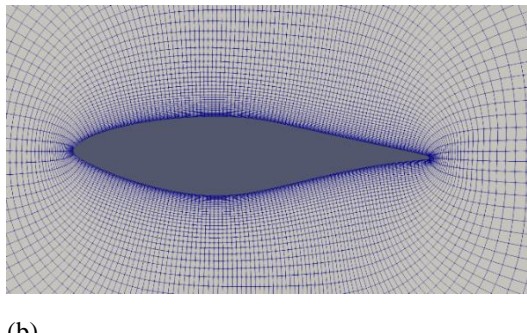

(a)                                                    (b)

**Figure 3. Mesh formulation: a) Farfield mesh layout; b) Hyperbolic expansion layer formulation**





To evaluate the mesh convergence and resolve the wake vortices with sufficient details and improve the accuracy of the CFD results, three levels of grids with varying grid densities are created while maintaining all other parameters constant. These

grids are called L0, L1, and L2, respectively. L0 has the finest mesh, L1 a medium mesh, and L2 a coarse mesh. The wind turbine's medium-sized mesh is seen in Figure 3(a). In order to better capture the vortices at the blade's tip and root, the mesh around those areas have been fine-tuned. There are sixty-five layers of boundary layer cells extruded from the turbine surface, as shown in Figure 3(b). The growth ratio is 1.2 and the initial cell height away from the turbine surface is set at 0.0005 m to keep the $y^+$ value below 10. $y^+$ is a non-dimensional distance (based on local cell fluid velocity) from the wall to

the first mesh node. To be confident in applying a wall function technique to a given turbulence model, $y+$ values must be guaranteed to fall inside a specified range. The blade pressure distribution was examined and converted to sectional aerodynamic force coefficient values to validate the accuracy of the CFD analysis. To simulate realistic turbulent flow around the blades and to validate the findings, wind tunnel test data is required. Although Reynolds-averaged Navier– Stokes equations (RANS) CFD analysis is more computationally expensive and CFD has limitations, it may be preferable to

simulate a wind tunnel test for examining surface pressure distribution. As for the turbulence model used, one-equation Spalart-Allmaras model is chosen due to its robustness, cheapness, good convergence, and fast implementation. For the whole, CFD analysis and optimization, a HPC with 72 processors is utilized.

Multi Reference Frame (MRF) is a steady-state technique used in Computational Fluid Dynamics (CFD) to describe issues with rotating components, therefore disregarding the tower and nacelle should have a minimal impact. It's less

computationally costly, yet it is still regarded precise enough. As mentioned above, three meshes are used for mesh convergence study, each with a progressively finer degree of detail, to ensure the accuracy of our estimations. Table 1 shows the details of the meshes used. The level 2 (L2) mesh is utilized in the design optimization given the computational cost of the optimization with L0 being used for design validation and analysis.

Pressure coefficients ($Cp$) can be computed with Eq. 1 along the blade, where $p$ is the local pressure on the blade surface and

$p_\infty$ is the atmospheric pressure. $u_\infty$ is the free-stream velocity and $\omega$ is the angular velocity.

$$C_p = \frac{p - p_\infty}{0.5\rho(u_\infty^2 + (\omega r)^2)} \tag{1}$$

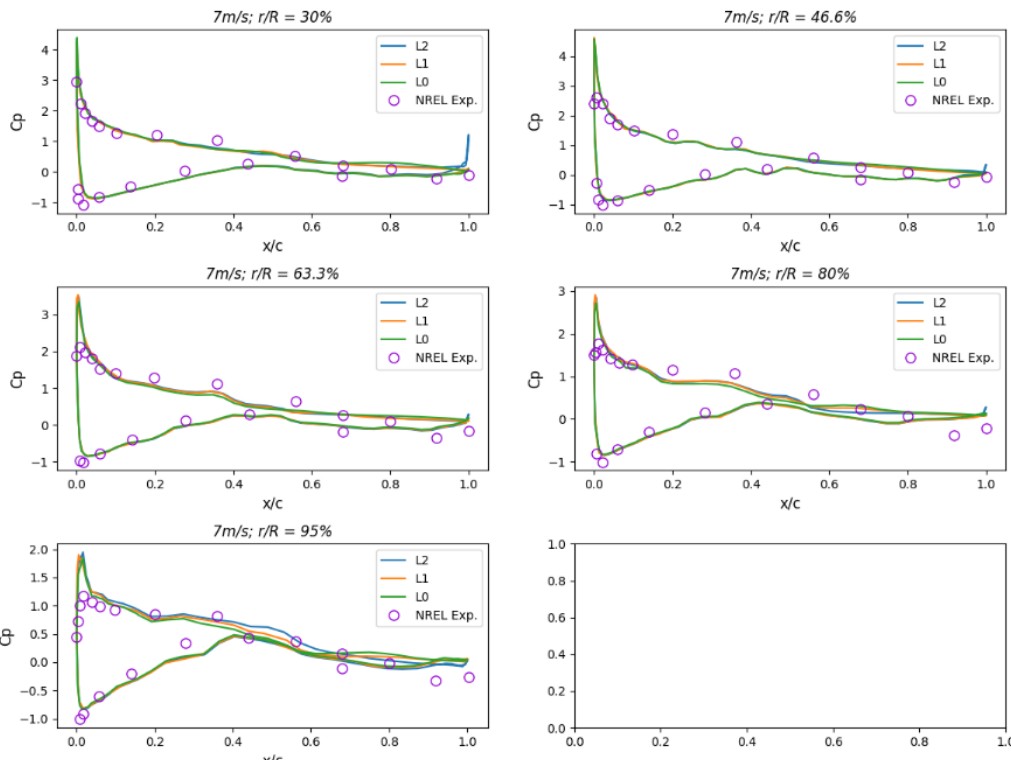

**Figure 4. Comparison of the pressure coefficients (*Cp*) from three levels of mesh (L2, L1, and L0) and NREL Exp.**

The torque and *Cp* vs *x/c* for a wind speed of 7 m/s are shown in Figure 4 to validate our CFD analysis against the
experimental data given by NREL (Simms et al., 2001). In Table 1, the torque data from various mesh levels are compared.
The results are sufficient to proceed with the optimization.  Although the mesh L0 is the most preferable and accurate one
among three sets of mesh levels, due to the computational cost, L2 is chosen for the optimization. Figure 5 depicts the
vorticity contour with the fine mesh L0, which shows uniform distribution, while the vorticity is still thick. Nevertheless, it is
good enough for RANS simulation to present that the mesh in the wake zone is sufficiently fine.

**Table 1. Mesh convergence by comparing the torque result from simulation against experimental value**

| Mesh# | Mesh type | Cells ($10^6$) | Torque (Nm) | Error (%) |
|---|---|---|---|---|
| L0 | Fine mesh | 17.857 | 712 | 9.2 |
| L1 | Medium mesh | 6.315 | 695 | 11.5 |
| L2 | Course mesh | 2.657 | 648.4 | 17.27 |
| NREL Exp. | - | - | 785 | - |





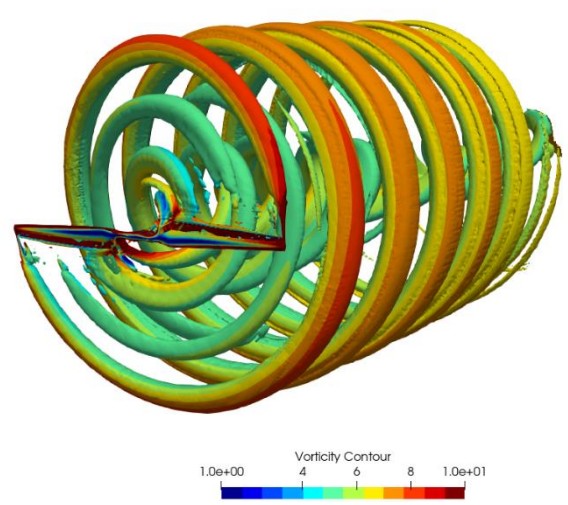


**Figure 5. Vorticity contour with L0 (fine mesh)**

### 3.2 Adjoint equations and optimization frameworks formulation

DAFoam implements the discrete adjoint solver using the FD Jacobian technique and automatic differentiation, which computes the partial derivatives using the coloring-accelerated finite difference method and solves the adjoint equations

using the Krylov method. The adjoint approach is used to quickly calculate the total derivatives $df / dx$, where $f$ denotes the goal or objective function, which is torque in our caseand $x$ is the vector of design variables. In the discrete method, it is assumed that the primal solver is capable of generating a discretized version of the governing equations and that the design variable vector $x \in R^{n_x}$ and the state variable vector $\omega \in R^{n_\omega}$ meet the discrete residual equations $R(\omega, x) = 0$, where $R \in R^{n_\omega}$ is the residual vector.

The relevant functions are thus functions of both the design and state variables: $f = R(\omega, x)$. While there are many functions of interest, in the following derivations, $f$ is treated as a scalar to maintain generality. As will become clear later, each new function necessitates the solution of another adjoint system. The chain rule is used to derive the total derivative $df / dx$:

$$\underset{1 \times n_x}{\frac{df}{dx}} = \underset{1 \times n_x}{\frac{\partial f}{\partial x}} + \underset{1 \times n_\omega}{\frac{\partial f}{\partial \omega}} \underset{n_\omega \times n_x}{\frac{d\omega}{dx}} \tag{2}$$



Where the partial derivatives $\partial f / \partial x$ and $\partial f / \partial \omega$ are reasonably straightforward to estimate due to the absence of implicit calculations. On the other hand, the total derivative matrix $d\omega / dx$ is costly since it is implicitly defined by the residual equations $R(\omega, \ x) = 0$.

Using the chain rule for R, we can get $d\omega / dx$. Because the governing equations should always hold, we can then leverage this knowledge to our advantage. As a result, the sum of the derivatives $dR / dx$ must be zero:

$$\frac{dR}{dx} = \frac{\partial R}{\partial x} + \frac{\partial R}{\partial \omega}\frac{d\omega}{dx} = 0 \Rightarrow \underbrace{\frac{d\omega}{dx}}_{n_\omega \times n_x} = -\underbrace{\frac{\partial R}{\partial \omega}^{-1}}_{n_\omega \times n_\omega}\underbrace{\frac{\partial R}{\partial x}}_{n_\omega \times n_x} \tag{3}$$

In Eq. (3), substituting $d\omega / dx$ from Eq. (2) yields:

$$\underbrace{\frac{df}{dx}}_{1 \times n_x} = \underbrace{\frac{\partial f}{\partial x}}_{1 \times n_x} - \overbrace{\underbrace{\frac{\partial f}{\partial \omega}}_{1 \times n_\omega}\underbrace{\frac{\partial R}{\partial \omega}^{-1}}_{n_\omega \times n_\omega}}^{\psi^T}\underbrace{\frac{\partial R}{\partial x}}_{n_\omega \times n_x} \tag{4}$$

Using $[\partial f / \partial \omega]^T$ as the right-hand side, we may solve the adjoint equation by transposing the state Jacobian matrix $\partial R / \partial \omega$.

$$\underbrace{\frac{\partial R}{\partial \omega}^T}_{n_\omega \times n_\omega}\underbrace{\psi}_{n_\omega \times 1} = \underbrace{\frac{\partial f}{\partial \omega}^T}_{n_\omega \times 1} \tag{5}$$

The $\psi$ is the adjoint vector. After solving this equation, we can get the total derivative by substituting the adjoint vector into Eq. (4), which results in:

$$\frac{df}{dx} = \frac{\partial f}{\partial x} - \psi^T\frac{\partial R}{\partial \omega} \tag{6}$$

For each function of interest, we need to solve the adjoint equations only once, because the design variable is not explicitly present in Eq. (5). Therefore, its computational cost is independent of the number of design variables, but it is proportional to the number of functions of interest. This approach is known as the adjoint method and is advantageous for many aerospace engineering design problems, in which we have only a few functions of interest but may use several hundred design variables.



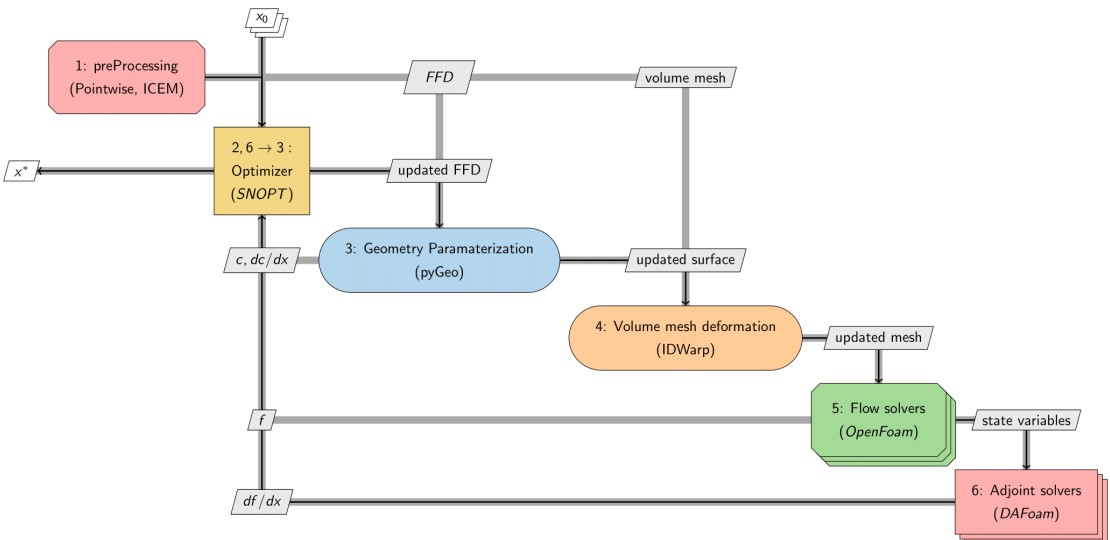

**Figure 6. Optimization framework shown by the extended design structure matrix (XDSM) (Lambe & Martins, 2012)**

Figure 6 represents the optimization framework, where the sequence in which components are performed is established by sequentially numbering them beginning with one. Numeric order is used to represent the sequence of execution. A thick gray line indicates data connections, while a thin black line indicates process connections. The modules are represented by the diagonal nodes, whereas the data is represented by the off-diagonal nodes. The stacked modules demonstrate parallel operation. The order in each node indicates the sequence of execution:

- In step 1 (preProcessing), a volume mesh is produced for the baseline geometry using Pointwise, which will be utilized later, as well as free-form deformation (FFD) points using ICEM, which will be used later in step 3 to morph the design surface (geometric parametrization).

- In step 2, the optimizer (SNOPT) is provided with a collection of baseline design variables. The design variables will be updated and be passed along to the geometry parameterization module (pyGeo).

- Step 3 accepts the updated design variables and FFD points created in the pre-processing phase, executes the deformation for the design surface, and then sends it to the mesh deformation module (IDWarp) for mesh deformation. In addition, pyGeo computes the values of geometric constraints and their derivatives with respect to the design variables.

- In step 4, IDWarp deforms the volume mesh in accordance with the modified design surface and delivers the
deformed volume mesh to the flow simulation module.

- In step 5, CFD tools (OpenFoam) are used in the flow simulation module to calculate the state variables in process 5 and send to adjoint solver.

- In step 6, the total derivatives of the objective function about design variables are computed and sent to the optimizer.



•    In the end, SNOPT gets the values and derivatives objective functions and constraints, executes the SQP computation, and returns a collection of updated design variables to pyGeo.

    •    The above procedure is repeated until the convergence is obtained.

### 3.3 Geometric parameterization and mesh deformation

The FFD control points are produced using ICEM and then parameterized using pyGeo. As such, FFD embeds the object's
geometry into a volume that may be changed by moving FFD points on the surface. The initial FFD box is constructed with volumetric control points, the design surface is embedded inside the FFD box, and the mapping between the design surface's physical coordinates and the FFD's parameter space is constructed. By repositioning the FFD control points, the design surface is distorted.

The number of FFD control points affects an optimization problem's design flexibility. Thus, the number of FFD control
points utilized influences the final design. Generally, 20 points are used for each airfoil segment while optimizing the design. As seen in Figure 7, the FFD box in this work is $10 \times 2 \times 15$, with $10 \times 2 \times 7$ for each blade and $10 \times 2 \times 1$ for the origin section. Ten control points are located on both the pressure and suction side of each segment throughout the span of each blade's seven sections, among which one section is fixed and the rest are free to be used for the optimization process.

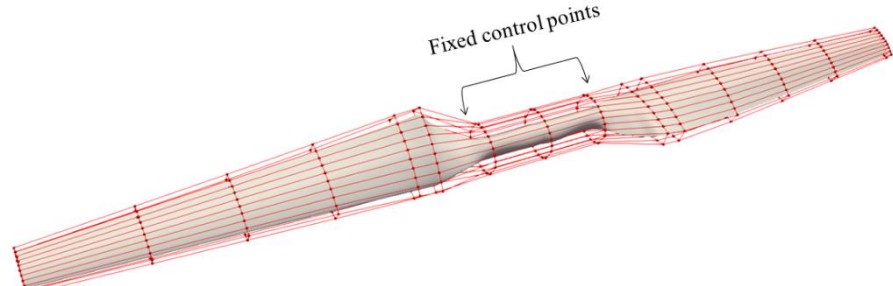

**Figure 7. FFD control boxes and points (red) for the optimization**

FFD points are utilized to optimize the local shape, twist, chord length, dihedral, and pitch angle of the blade. Due to the fact that the blades must be optimized equally and concurrently, they are compelled to undergo the same amount of change due to symmetrical constraints. Three stations are fixed between two blades. Concerning the remaining geometric constraints, volume and thickness constraints are applied, while LE and TE constraints are added to fix the leading and trailing edges and
allow for simultaneous local shape and twist optimization. The reference axis is specified directly by passing DVGeometry a pySpline curve object at 30% from the leading edge.

Pitch optimization requires that all portions twist the same amount, while twist optimization requires that each segment twist separately. During the chord length optimization process, only the chords of the root and tip parts are optimized, and the areas in between are interpolated. In the case of the dihedral, the displacement in the $y$ direction relative to the reference axis
is added to the control points. During the entire shape optimization process, shape optimization in the $y$ direction is



performed in conjunction with twist, chord length, and dihedral. Constraints on the leading and trailing edges are imposed to prevent the local shape variables from causing a shearing twist.

The lelist and telist points (10 × 15) are used for each blade from the root to the tip. Namely, 10 constraints points from the leading edge to the trailing edge at the 15 sections of the blade spanwise. They totally lie inside the blade and used for the thickness and volume control points.

### 3.4 Design Optimization algorithm formulation

In our study, the merit function to maximize is the torque in $y$ direction with respect to pitch, planform, and full shape optimization. The freestream velocity $u_\infty = 7$ m/s while the rotational velocity $\omega = 72$ rpm.

**Table 2. Optimization algorithms**

| Design variables | Number of design variables | Optimization schemes | | | | |
|---|---|---|---|---|---|---|
| | | S1 | S2 | S3 | S4 | S5 |
| Pitch | 1 | ✓ | | | | |
| Shape | 120 | | ✓ | ✓ | ✓ | |
| Twist | 6 | | ✓ | | | ✓ |
| Chord | 2 | | | ✓ | | ✓ |
| Dihedral | 6 | | | | ✓ | ✓ |
| Total | | 1 | 126 | 122 | 126 | 14 |

As shown in Table 2, the optimization algorithms are categorized as follows and the optimization is conducted accordingly.

- Optimization scheme S1: pitch optimization with pitch angle as a single design variable
- Optimization scheme S2: shape and twist optimization with 120 shapes in $y$ direction and 6 twist angles as design variables (126 in total)
- Optimization scheme S3: shape and chord optimization with 120 shapes in $y$ direction and 2 chord lengths as design variables (122 in total)
- Optimization scheme S4: shape and dihedral optimization with 120 shapes in $y$ direction and 6 dihedrals in $y$ direction as design variables (126 in total)
- Optimization scheme S5: twist, chord, and dihedral optimization with 6 twist angles, 2 chord lengths and 6 dihedrals in $y$ direction as design variables (14 in total)





## 4.    Results and discussion

The optimization with mesh L2 is implemented and the results are presented and discussed below. The optimizations with
respect to such optimization schemes as S1(pitch optimization), S2(shape and twist optimization), S3(shape and chord
optimization), S4(shape and dihedral optimization) and S5(twist, chord and dihedral optimization) are carried out
respectively and compared respecting the pre and post optimization results.

### 4.1 S1: pitch optimization

During the pitch angle optimization, the FFD points at the six sections along the blade (six twist angles) are controlled as a
single design variable to get the result shown in Figure 8 and 9.

Figure 8 depicts the improvement in the merit function as well as the change in optimality in terms of S1 (pitch), which was
achieved after only four iterations. Despite the fact that the improvement in torque is 5.34 %, the optimality is less than
0.0002, while the optimality tolerance is 1E-7.

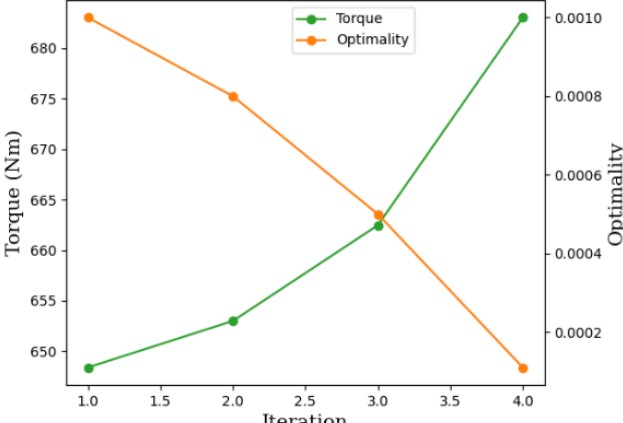

**Figure 8. Optimization process with respect to S1 (pitch)**

The optimization is achieved by changing the pitch angle from 0 degrees to 4.91°, as illustrated in Figure 9. As previously
noted, the increase in torque ranges from 648 Nm to 683 Nm, representing a 5.34% increase.





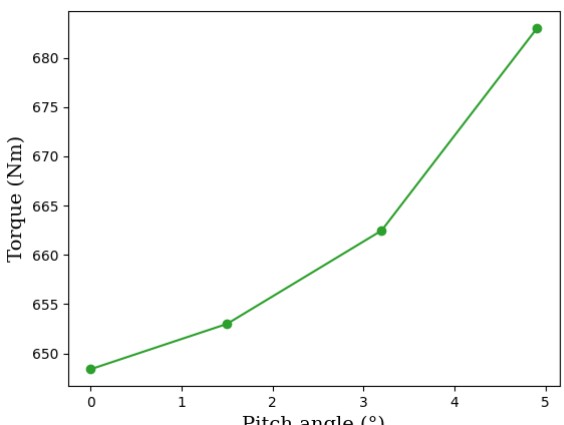

**Figure 9. Merit function improvement in terms of S1 (pitch angle)**

And the optimization is performed in S1 (pitch angle) mode, which results in a simultaneous change in twist along the blade
of the same amount, which can be up to 4.91° as shown in Figure 10. The improvement takes place between the stations from
1.25 m to the tip. Pitch angle is optimized by changing the twist of all the sections along the blade with the same amount
simultaneously, which was mentioned in the methodology section.

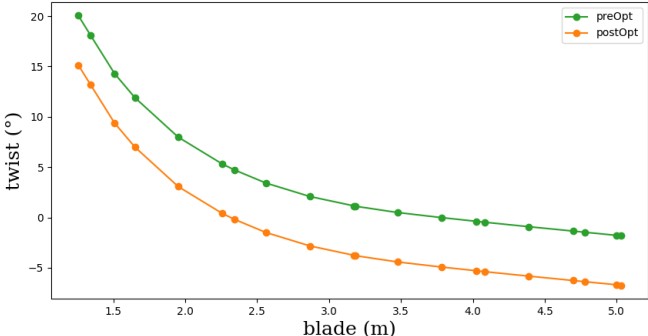

**Figure 10. Twist angle variation with respect to S1 (pitch angle)**

**4.2 S2: shape and twist optimization**

After the pitch angle optimization in S1, the combination of the shape and twist is considered as in S2 for the optimization.
In this optimization, 120 (10 x 2 x 6) control points at the six stations along each blade for shape and 6 sections of FFD
points as design variables along each blade are used. And the corresponding result of the optimization is presented in Figure
11 in terms of the objective function and in Figure 12, the pressure coefficients (*Cp*) before and after the optimization is
compared.





Figure 11 presents the merit function improvement as well as the optimality change in terms of S2 (shape and twist), which converged in 10 iterations. The improvement in torque is 23.26 % while the optimality reached below 0.01. However, the convergence optimality tolerance for the optimization is 1E-7, which is much lower than the optimality which occurred for the optimization in our case.

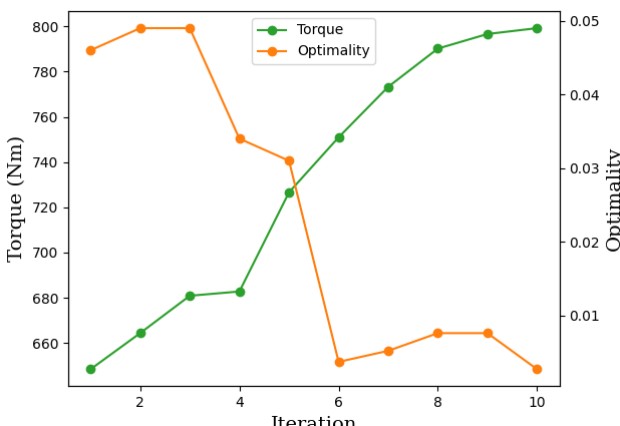

**Figure 11. Optimization process with respect to S2 (shape and twist)**

During the optimization, the pressure coefficients ($Cp$) before and after the optimization at three stations at 30%, 63% and 95% along the blades are compared in Figure 12, while the pressure distribution on the blade before and after the optimization is presented as well.

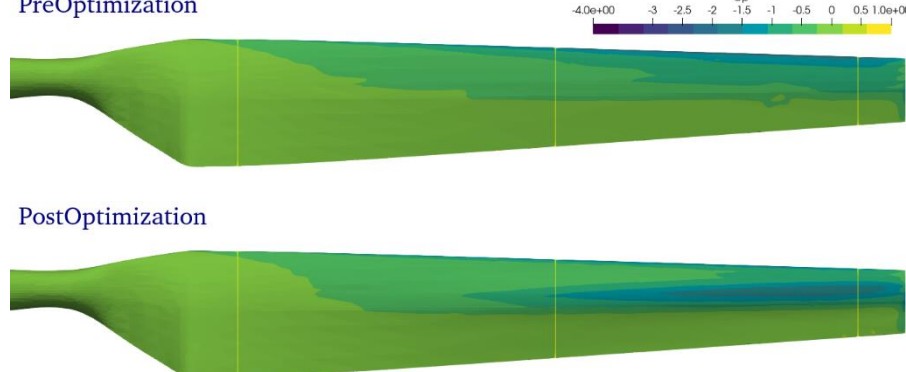





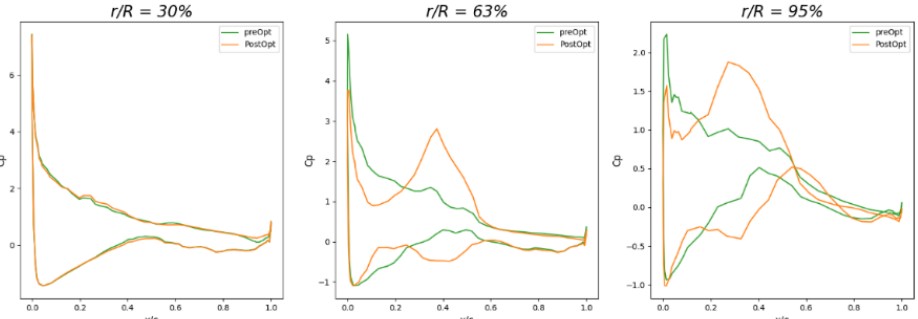

**Figure 12. Comparison between the baseline and optimized blade in terms of S2 (shape and twist)**

The pressure coefficients (*Cp*) and the pressure change close to the blade tip (*r/R = 95%*) is considerable compared to the ones near the root (*r/R = 30%*) as shown in Figure 12. In optimization scheme S2, both the shape and the twist are improved to lead to the improvement in the merit function and the twist variation along the blade is plotted in orange while the green one presents the twist before the optimization. The twist close to the tip changes substantially compared to the root and the variation is seen by the difference of the twist angle values before and after the optimization in Figure 13. As sections are optimized in terms of twist along the blade and the sections between them are interpolated accordingly, the location of the six sections is marked in orange on the optimized blade twist profile while unoptimized twist profile shows twenty marked stations in green along the blade.

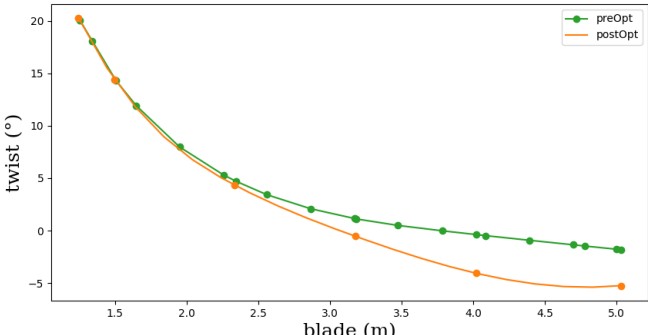

**Figure 13. Twist angle variation with respect to S2 (shape and twist)**

### 4.3 S3: shape and chord optimization

After the optimization S1(pitch angle) and S2(shape and twist), the combination of the shape and chord are taken into the consideration in S3. During this optimization, 120(10 x 2 x 6) control points for the shape at the six stations along each blade and 2 sections of FFD points as 2 design variables at tip as well as at the root along the blade are utilized.

Rather than optimizing shape and twist, in optimization scheme S3 the shape and chord length is improved to maximize the objective function torque in the *y* direction in terms of 120 variables for shape and 2 variables for the chord length. Optimization history in terms of the merit function variation and optimality with respect to the iteration sequence is plotted




in Figure 14 while pressure on the suction side of the blade as well as the pressure coefficient before and after the improvement at the three sections (30%, 63%, 95%) shown are compared in Figure 15.

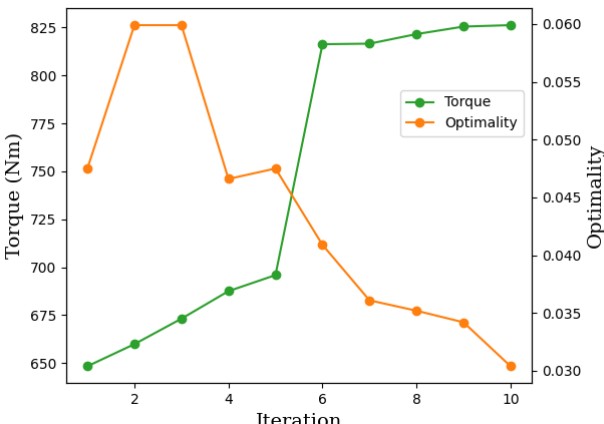

**Figure 14. Optimization process with respect to S3 (shape and chord)**

It takes 10 iterations to reach the optimization convergence in this case as seen in Figure 14, where optimality decreases to about 0.03 and the torque increases from 648 Nm to 823 Nm, which is as high as 27%. At the end of the fifth iteration, there

takes place a sharp growth of approximately 100 Nm in the torque and then grows gently till the end of the optimization at the iteration 10.

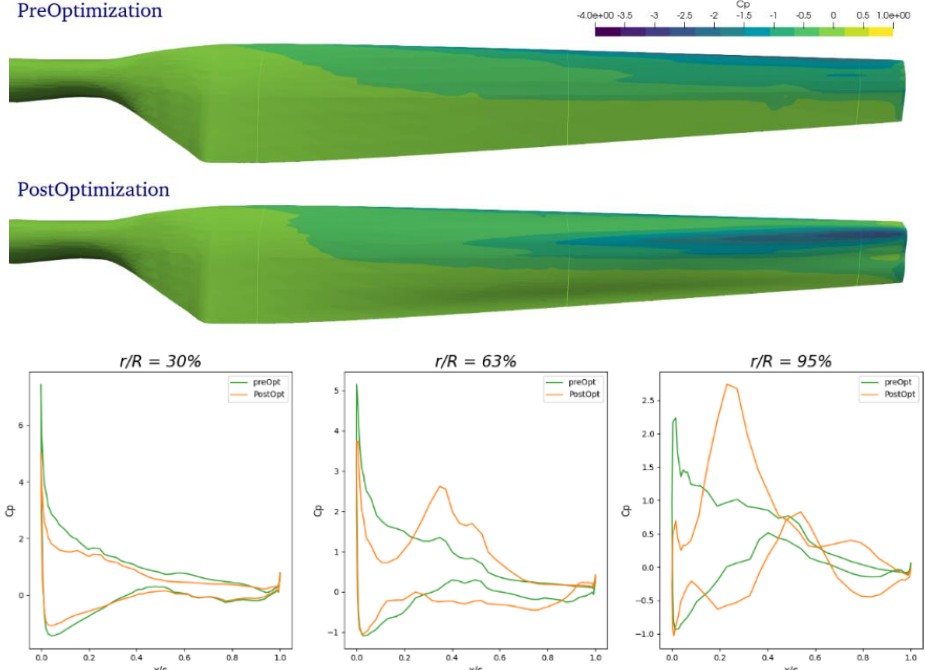

**Figure 15. Comparison between the baseline and optimized blade in terms of S3 (shape and chord)**



Comparing the pressure distribution and the *Cp* value of the pre-optimized and post-optimized blade as presented in Figure 15, one can easily spot the abrupt change in the values of *Cp* at the sections close to the tip of the wind turbine blade which

is the main contributor to the power capture. And the change in the pressure and the pressure coefficient (*Cp*) in the vicinity of the blade tip also accounts for the optimized value of the torque. On the contrary, the changes in the blade root are not as considerable as the ones near the tip. As mentioned earlier, two sections (one at the root and another the tip) are chosen for the chord optimization and thus two design variables for the chord length are varied to obtain the optimum value. The sections between the two sections optimized can be obtained by the interpolation.

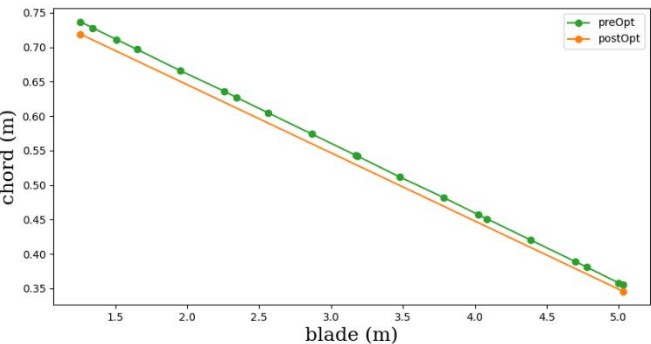


**Figure 16. Chord length variation with respect to S3 (shape and chord)**

Observing Figure 16, the chord length reduces both at the root and the tip so does at the sections between. The reduction in the chord length is around 2.5%.

### 4.4 S4: shape and dihedral optimization

In the wake of considering the optimization in terms of the combination of shape and twist, shape and dihedral as in S2 and S3, the combination of shape and dihedral in S4 is examined, and thisscheme (S4) consists of 126 design variables in which 120 for the shape in *y* direction and 6 for the dihedral. The optimization converged with the optimality just below 0.024 and the optimized torque as much as above 700 Nm and 8.7% of the optimization with respect to the objective function, torque, is procured as illustrated in Figure 17.





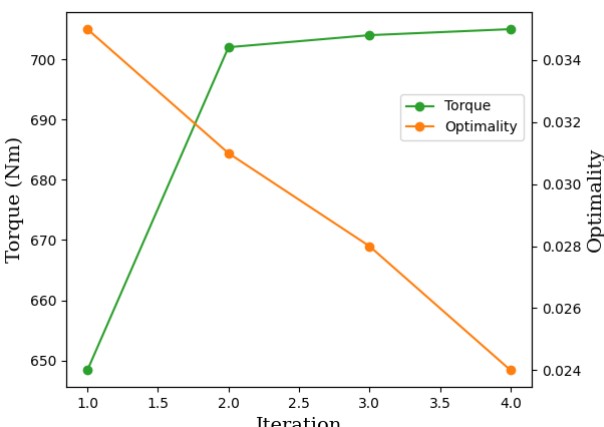


**Figure 17. Optimization process with respect to S4 (shape and dihedral)**

While the optimization in the objective function is depicted as in Figure 17, in Figure 18 the pressure coefficients (*Cp*) at the three stations (30%, 63% and 95%) are compared with respect to the pre-optimized and post-optimized values. One can also find the pressure changes on the blade surface after the optimization. Most of the changes in the pressure and pressure

coefficients occur at the leading edge near the tip of the blade.

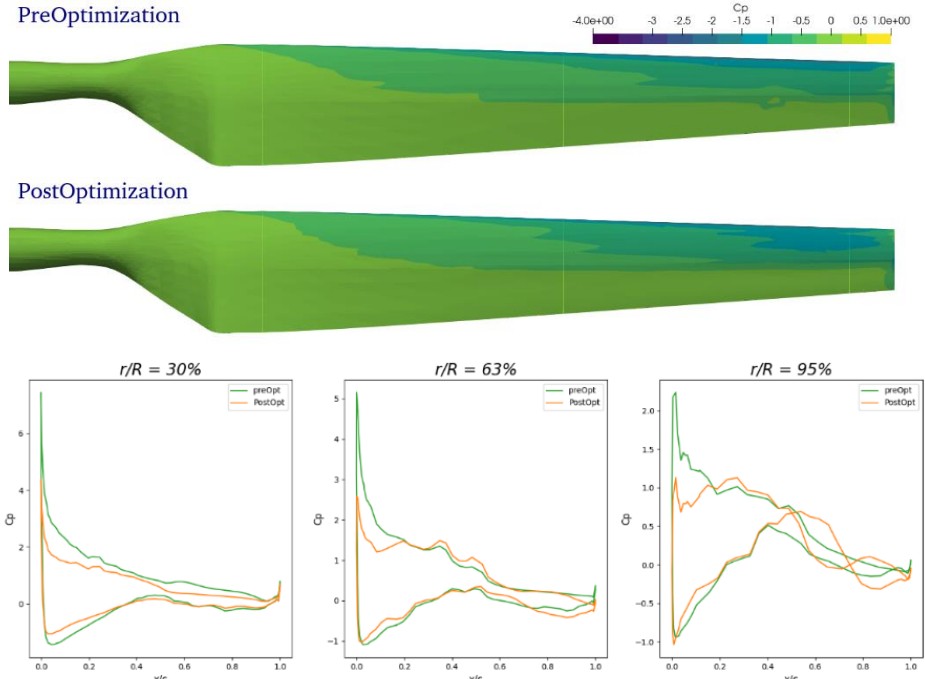

**Figure 18. Comparison between the baseline and optimized blade in terms of S4 (shape and dihedral)**





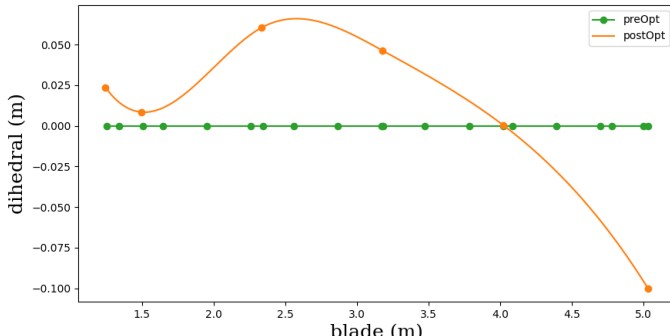

**Figure 19. Dihedral variation with respect to S4 (shape and dihedral)**

Meanwhile the dihedral improvement can be found in Figure 19 as only six sections are optimized with the corresponding

control points, which are marked in orange. Before the optimization, the dihedral value along the blade is zero. The change

in the dihedral along the blade is around a few centimeters at the end of the optimization. As only six points are obtained

through the dihedral optimization, the rest of the parts are interpolated accordingly. Therefore, the smooth curve is gained as

shown in orange in Figure 19. The dihedral change might be somewhat illogical partially because of the limited

computational capabilities of the HPC.

**4.5 S5: twist, chord and dihedral**

While in the above sections the combination of the shape and twist, chord length and dihedral have been taken as the design

variables respectively, in this section the combination of twist, chord lengths and dihedral are considered for the

optimization.

Usually, twist and chord optimizations are called planform optimization and the planform optimization for the wind turbine

blades has been conducted by other researchers without considering dihedral. In this study, dihedral optimization at the 6

sections is included in the planform optimization apart from the chord length and twist angle optimization. And in this work,

it is categorized as optimization scheme S5, which has 14 design variables.





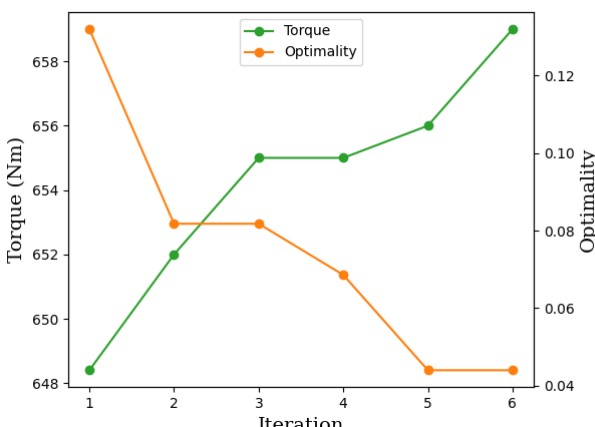

**Figure 20. Optimization process with respect to S5 (twist, chord and dihedral)**

Demonstrated in Figure 20, due to the optimization scheme S5, merely 1.65% increment come about, which is substantially lower in comparison to the optimized values with other schemes discussed above. With this scheme, it takes 6 iterations to hit the maximum point of the merit function. And the optimality comes to spot slightly above 0.04.

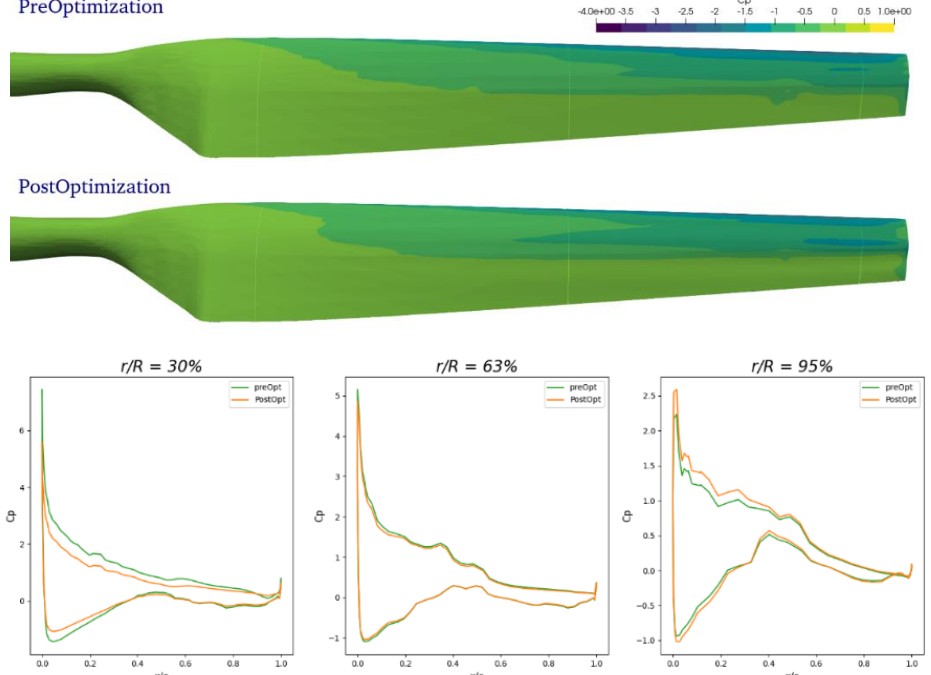

**Figure 21. Comparison between the baseline and optimized blade in terms of S5 (twist, chord and dihedral)**

Considering the pressure coefficients (*Cp*) at three sections (30%, 63%, 95%) on the blade prior to as well as subsequent to

the optimization in Figure 21, the alteration is more significant on the surface nearby both ends while the change isn't very





obvious in the middle. Nevertheless, taking into consideration the pressure distribution on the unoptimized as well as optimized blade surfaces, the main variation happens at the leading edge of the blade.

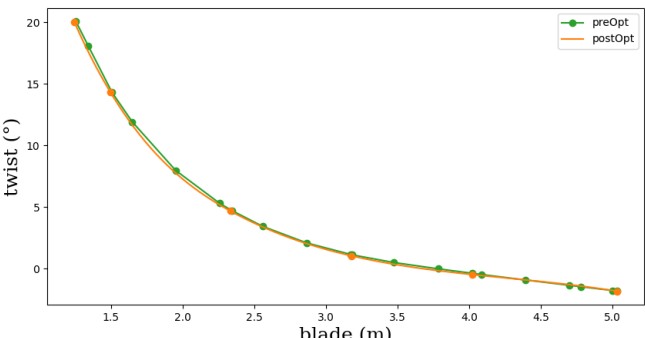

**Figure 22. Twist variation with respect to S5 (twist, chord and dihedral))**

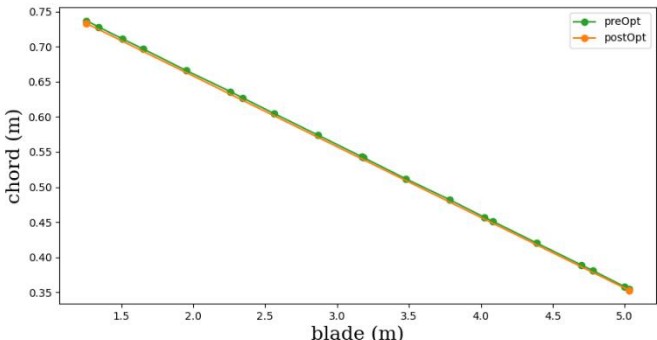


**Figure 23. Chord length variation with respect to S5 (twist, chord and dihedral)**

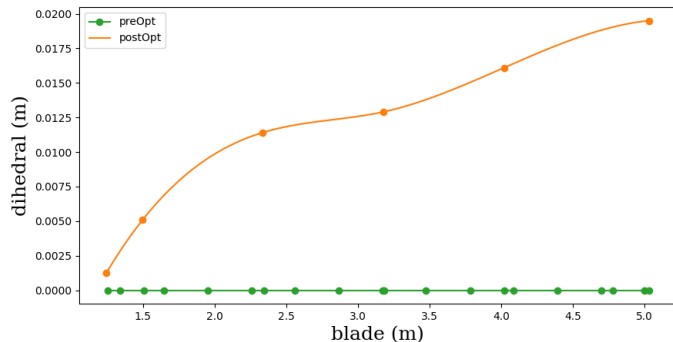

**Figure 24. Dihedral variation with respect to S5 (twist, chord and dihedral)**

As abovementioned, the 14 design variables for the twist angles, chord lengths as well as the dihedral are taken into account

in optimization scheme S5. Examining the Figures 22-24, the crucial change is in the dihedral in contrast to the other factors




such as twist and chord lengths which alter marginally to play a part in the optimization of the objective function. Observing the Figure 22 and Figure 23, the chord and twist undergo a minor difference compared to the baseline. Nevertheless, given the results from the optimization schemes S2 and S3, notable augmentation is attained with the combination of the shape and twist, as well as the shape and chord length.

**5.     Conclusion**

The aim of this study: to contribute towards the development of an open source MDO (multidisciplinary design optimization) platform DAFoam in general and develop, implement and integrate various technologies for wind turbine MDO for the energy community in particular. This work represents the first attempt to implement DAFoam for wind turbine optimization; to implement high-fidelity aerodynamic design optimization for the wind turbine blades to increase the power

output generated in terms of five different schemes.

In this paper RANS-based high-fidelity concurrent aerodynamic optimization with discrete adjoint method has been successfully demonstrated on the NREL phase VI in terms of five different optimization schemes such as S1, S2, S3, S4 and S5, and the results are illustrated and analyzed in the results and discussion section. Three levels of meshes are generated, and verification and validation are performed against the experimental results. Mesh L2 (course mesh/2.6 million) is selected

and employed for the optimization due to the high computational cost of finer meshes, while mesh L0 is used for design validation and aerodynamic analysis.

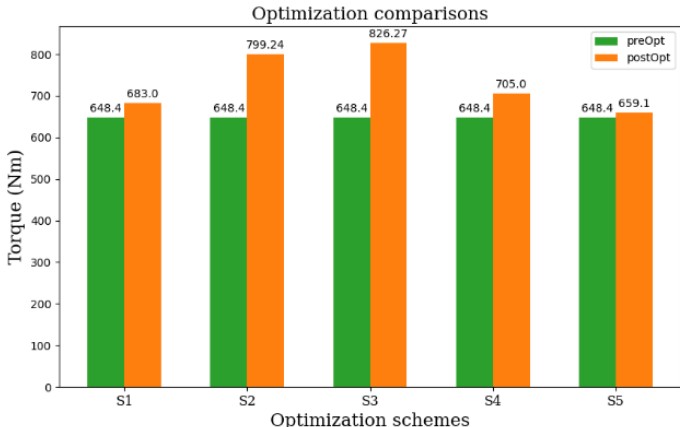

**Figure 25. Merit function before and after the optimization**

As a conclusion, the merit function values before and after the enhancement are compared in Figure 25 with respect to the

five optimization methods, where the values prior to the optimization are plotted in orange while the ones subsequent to the optimization are plotted in green. The most significant improvement takes place in the optimization schemes S2 and S3, namely in which the combination of shape and twist, shape and chord was optimized. Therefore, it can be concluded that the



most important aerodynamic parameters to be optimized during the wind turbine blade optimization are the combination of cross-sectional shape, twist and chord.

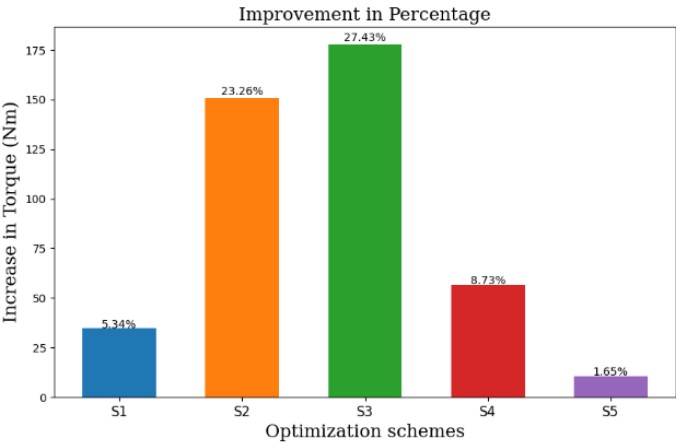


**Figure 26. Improvement in the torque by percentage**

Again as demonstrated in Figure 26, the schemes S2 and S3 bring about the most remarkable augmentation in the torque whereas the insignificant increment takes place with the case of scheme S5. Since the principal focus of this work is the shape and planform optimization with as many as 135 design variables in all, multidisciplinary design optimization isn't 515 taken into account. As a continuation of this work, multidisciplinary design optimization will be pursued with the addition of the solid solver Solid4foam and TACS, VLES (Very Large Eddy Simulation) solver (Maulenkul et al., 2021) and fluid-structure interaction methods being implemented in (Dinmukhamed Zhangaskanov et al., 2022).

**Data availability.**

Data are available upon request to the first author (shaheidula.batai@nu.edu.kz)

**Author contribution.**

Yong Zhao oversees the project and provides the fund as well as the equipments; Sagidolla Batay developed the geometry, CFD analysis and optimization process and wrote the manuscript. Bagdaulet Kamalov and Dinmukhamed Zhangaskanov assisted with the mesh generation and manuscript. Yong Zhao, Dongming Wei, Tongming Zhou and Xiaohui Su reviewed 525 and edited the manuscript.





**Competing interests.**

The authors declare that they have no conflict of interest.

**Acknowledgement and funding.**

This study is funded by Nazarbayev University through a FDCR grant No. 240919FD3934

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
