# Peer review of "Adjoint-Based High Fidelity Concurrent Aerodynamic Design Optimization of Wind Turbine"

_Wind Energy Science, 2022_

## Referee Comment (RC1)

**Review of "Adjoint-Based High Fidelity Concurrent Aerodynamic Design Optimization of Wind Turbine" by S. Batay et al.**

The authors present an optimization study of a wind turbine blade based on the DAFoam open source toolbox in OpenFOAM. My main concern is the limited novelty of the current paper. As it is written, the manuscript is a rather straightforward application of existing software to a case for which a tutorial exists on the software website. Furthermore, there is significant room for further improvement on detailing the setup for reproducibility, improving figure quality, and enhancing overall paper readability. My most important comments are detailed below.

**Major comments**

1. The authors highlight the novelty of their work as being "the first ever work on wind turbine optimization implementing and using the DAFoam to data based on adjoint and RANS solvers with reverse AD technique". A first thorough application of a widely available DA framework to wind turbine design holds value, but I believe this statement illustrates the marginal novelty of the current work. The authors cite some papers where discrete adjoint optimization is performed on the same NREL VI turbine as considered here (L166), and the DAFoam website even features a tutorial for the NREL VI wind turbine case: https://dafoam.github.io/mydoc_tutorials_aero_nrel6.html . As such, I find the claim somewhat exaggerated. A documented application of a software tutorial does not warrant a journal publication. The authors should therefore much more clearly outline how their manuscript advances the state of the art.

   Furthermore, in the conclusion, the authors claim the aim of this study is "to contribute towards the development of an open source MDO platform", however, I see no development in the current work, but rather an application to a specific case.

2. The CFD setup is insufficiently detailed:
   a. Figure 3a: Can the wind turbine geometry be included in this figure, or in an overall illustration of the entire domain. How large is the domain with respect to the turbine geometry?
   b. Boundary conditions should be discussed, this is missing
   c. The description of y+ in line 229, based on a local cell fluid velocity, is confusing, if not completely incorrect.
   d. It is not clear whether a wall function is applied and, if yes, which one.
   e. L237: It seems irrelevant to mention "HPC with 72 processors" without clarifying which processor was used and detailing computational aspects such as wall time or memory usage.
   f. L232: "Although Reynolds-averaged Navier– Stokes equations (RANS) CFD analysis is more computationally expensive and CFD has limitations, it may be preferable to simulate a wind tunnel test for examining surface pressure distribution." I'm having trouble understanding the relevance and intention of including this sentence.
   g. It is not completely clear whether the simulation performed is a steady RANS or a URANS. L238 makes me suspect the former, but this should be more clearly mentioned earlier in the setup description.

      h.    Which OpenFOAM solver is used? I suspect simpleFoam, but this should be explicitly mentioned.

3.   The discussion of the adjoint equations seems unnecessary and is formulated a little too informal.
      a.    L268: R is used both for a vector space of real numbers, as well as for the definition of the residual. Also, the vector space dimensionalities n_x and n_\omega are not explicitly defined
      b.    The train of thought to arrive at Equation 5 is not clear
      c.    Overall, the adjoint derivation adds very little to the narrative other than the statement that 'The computational cost of the gradient computation is independent of the number of design variables'. I would suggest to remove the derivation and instead refer to literature for a more thorough and rigorous explanation.

4.   The manuscript is lacking adequate references to literature in key places where the authors explicitly mention prior research has been conducted, for example
      a.    L48 The sentence mentions "in certain research" without referring to such works. Overall, more references should be added here.
      b.    L65 "While the majority of studies investigate …., others explore the combination of the two" References should be added.
      c.    L132: "As a result, their applications have often been limited to …" Add at least 1 reference per example

5.   The authors use a very cumbersome style of incorporating references in their sentences which impedes fluent readability of the manuscript. Surely this can be formulated in a less wordy manner. Examples:
      a.    L83: Ning (Andrew Ning, 2013) addressed this …
      b.    L86: Chehouri et al. (Chehouri et al., 2015) conducated an in-depth …
      c.    L142: Peter and Dwight (Peter & Dwight, 2010) discussed these …
      d.    L143: Martins and Hwang (Martins & Hwang, 2013) expanded …
      e.    L166: T. Dhert et al. (Dhert et al., 2017) and Mads H. Aa. Madsen et al. (Madsen et al., 2019) have recently …

6.   The writing structure and style of the manuscript should be improved overall, the manuscript could be significantly streamlined to improve readability:
      a.    L39 and L57 deliver an identical message, please revise and avoid unnecessary repetition.
      b.    L73 "Wind turbine performance (power production and loads) has been shown to be affected by the aerodynamic forces created by the wind" This is a trivial statement, can be omitted easily.
      c.    Section 2.2 adds very little to the overall narrative of the manuscript and could be omitted. Also, in the final line where computational expense is discussed, citing a paper from 1952 seems outdated.
      d.    Section 2.3: categorizing RANS CFD as high fidelity should be done with caution. Perhaps the authors should refer to RANS techniques as CFD-based methods rather than high fidelity, as there is significant uncertainty regarding turbulence modeling

e. Section 2.3.1, is there need to make a separate sub-subsection for this topic, if there is no distinguished counterpart 2.3.2? Could be just a paragraph rather than a sub-subsection.

f. Words are often capitalized unnecessarily, e.g.
   i. Section 2.3 title: why is Gradient capitalized?
   ii. L128: genetic algorithms, simulated annealing, particle swarm optimization should not be capitalized
   iii. L152: partial differential equations

7. The quality of the figures should be improved significantly:
   a. The resolution of most figures is too low for publication in a journal article. Wherever possible, I would suggest to use vector image formats or higher resolution rasterized figures. For example, Figure 4 is perfectly suitable for a vectorized format.
   b. Figure layout and font size should be improved for almost every figure. A non-exhaustive list of corrections to be made:
      i. Figure 4: The 3 x 2 layout with an empty subplot is awkwardly formatted with the bottom right empty axis. I recommend another subplot layout, or at least remove the empty axis entirely.
      ii. Figure 5: font size in colorbar text is too small
      iii. Figure 6: font size is too small overall
      iv. Figure 8: update the x ticks to only contain integers, it does not make sense to label half iterations. Also, make your figure self-contained. The label "Optimality" by itself does not mean anything.
      v. Figure 12: fonts way too small
      vi. Many figures are quite sparse in terms of information density, and hence take up a lot of space relative to the information presented. Figures could easily be combined to make the paper more compact and concise, e.g. Figure 9 and 10 could perfectly be placed side-by-side

**Minor comments**

1. Abstract, first sentence is very long, complex, and convoluted. Simplifying would benefit readability
2. L184: Can the authors clarify what the reverse AD technique is?
3. L252: "Figure 5 depicts the vorticity contour with the fine mesh L0, which shows uniform distribution, while the vorticity is still thick. Nevertheless, it is good enough for RANS simulation to present that the mesh in the wake zone is sufficiently fine." What is meant by 'the vorticity is still thick'? How does this justify the mesh resolution?
4. L374: "the optimality is less than 0.0002". 'optimality' is not an unambiguously accepted term. Please provide further detail on what is meant here. Same for optimality tolerance. Are these related to a gradient norm?
5. Figure 9 seems superfluous, it shows the same exact green line as Figure 8, but with a different axis. Please merge both figures.
6. L459: What is meant by "The dihedral change might be somewhat illogical partially because of the limited computational capabilities of the HPC"
7. L491: "The aim of this study: to contribute towards …" please write in full sentences

**Technical / typographical corrections**

1. L14: hi-fidelity -> high-fidelity
2. L28: "excellent results are achieved" -> non-descriptive and somewhat suggestive, please reformulate in a quantifiable manner
3. L43: "for its improving" -> for improving
4. L46: Fast (should be FAST) and Open Fast (should be OpenFAST) are essentially the same software. Also, it would make sense to add a reference per code.
5. L94: abbreviation MDF is not used anywhere else in the manuscript, can be omitted. Same for SA and PSA in L129
6. L115: Navier Stokes -> Navier-Stokes
7. L151 & L158: PDE is introduced twice
8. L129: GA is introduced twice
9. L151: do not capitalize partial differential equations
10. L152: "performing an accurate differentiation of the system of PDEs is quite difficult" This is a very vague statement, especially since it's followed by a discussion of the adjoint technique which allows gradients to be computed with relative ease. Please refine this statement.
11. L184: AD is undefined
12. L234: RANS is introduced here, but is already used much earlier multiple times in the manuscript
13. L249: "The torque and Cp vs x/c for" don't use 'vs', please use full sentences.
14. Use $C_P$ instead of Cp throughout manuscript
15. L263: FD is not defined